# Using Artificial Intelligence Techniques to Predict Punching Shear Capacity of Lightweight Concrete Slabs

**DOI:** 10.3390/ma15082732

**Published:** 2022-04-07

**Authors:** Ahmed Ebid, Ahmed Deifalla

**Affiliations:** Department of Structural Engineering and Construction Management, Future University in Egypt, New Cairo 11835, Egypt; ahmed.abdelkhaleq@fue.edu.eg

**Keywords:** punching shear, lightweight, GP, EPR, ANN

## Abstract

Although lightweight concrete is implemented in many mega projects to reduce the cost and improve the project’s economic aspect, research studies focus on investigating conventional normal-weight concrete. In addition, the punching shear failure of concrete slabs is dangerous and calls for precise and consistent prediction models. Thus, this current study investigates the prediction of the punching shear strength of lightweight concrete slabs. First, an extensive experimental database for lightweight concrete slabs tested under punching shear loading is gathered. Then, effective parameters are determined by applying the principles of statistical methods, namely, concrete density, columns dimensions, slab effective depth, concrete strength, flexure reinforcement ratio, and steel yield stress. Next, the manuscript presented three artificial intelligence models, which are genetic programming (GP), artificial neural network (ANN) and evolutionary polynomial regression (EPR). In addition, it provided guidance for future design code development, where the importance of each variable on the strength was identified. Moreover, it provided an expression showing the complicated inter-relation between affective variables. The novelty lies in developing three proposed models for the punching capacity of lightweight concrete slabs using three different (AI) techniques capable of accurately predicting the strength compared to the experimental database

## 1. Introduction

Devastating casualties could result from a concrete slab’s failure under punching shear, which is due to it occurring suddenly with no warning. Many recent punching shear failures have been reported worldwide [1,2,3,4,5]. Several design models were developed for punching shear strength; however, these models vary significantly in the considered parameters and mechanism in developing the model [6,7,8]. For example, the European concrete design code (EC2) [6] model is semi-empirical. In contrast, the FIB model design code (MC) [7] is physically based. Thus, punching shear strength continues to be a dilemma that involves many parameters and mechanisms affecting the strength [9,10,11,12,13,14]. The load transfer mechanisms for punching shear strength consists of the following components: (1) the direct shear resistance of the un-cracked compression zone; (2) the friction shear across the diagonal crack from the aggregate interlock and interface shear; (3) the bending and direct shear resistance of the flexure reinforcement crossing the cracks (i.e., dowel action). The effective parameters are as follows: (1) yield strength of steel; (2) concrete properties; (3) flexure reinforcements; (4) slab dimensions; and (5) column dimensions.

The cost of slabs is relatively high; thus, the engineering community tends to look at reducing the cost via implementing new materials, such as concrete with fiber-reinforced polymer, lightweight concrete (LWC), fiber-reinforced concrete, or high-strength concrete [15,16,17,18,19,20]. However, LWC is an overwhelming choice, different from conventional normal-weight concrete (NWC) in the aggregate type. In NWC, diagonal shear cracks propagate around the strong aggregates; thus, sharp aggregate sticks out across the crack edges. On the other hand, diagonal shear cracks in LWC will go through the weak aggregate [21,22,23]. Therefore, the LWC punching shear strength could be different from that of NWC.

For the case of LWC slabs under punching shear, design codes are based on research studies focused on investigating either LWC beams under one-way shear [23,24] or NWC slabs under punching shear [25,26,27,28,29]. In addition, there are very limited studies in the available literature verifying the most recent revision of the American concrete design code (ACI) [8] design code for the punching shear of LWC slabs. Globally, researchers have tackled the punching shear design of NWC and LWC slabs for more than six decades. Most of the research studies were experimental ones [30,31,32,33,34,35,36,37,38,39,40,41,42,43,44,45]. Based on this pioneering work of Moe [30], the ACI [8] punching shear design provisions for NWC slabs were developed. Later, based on the pioneering work of Hognestad [31] and Ivy [32], the ACI was modified to fit LWC slabs. Those modifications used either the actual tensile splitting strength or a reduction factor (λ) for calculating the shear strength of LWC slabs. A more recent study by Caratelli [39] concluded that the MC is conservative for the case of LWC under punching shear; thus, increasing the aggregate factor by 40–60% of the nominal maximum aggregate size was proposed. On the other hand, the ACI [8] proposed a modified reduction factor using the concrete density, which was developed to investigate LWC beams under one-way shear [22,23].

This current study investigates the punching shear strength of LWC slabs. An extensive experimental database for LWC slabs tested under punching shear loading is collected. Effective parameters were selected using statistical analysis: concrete density, columns dimensions, slab effective depth, concrete strength, flexure reinforcement ratio, and steel yield stress. Three artificial intelligence (AI) techniques were selected and used to develop punching shear strength models for LWC slabs. Strength predictions were compared with each other and with selected design codes concerning the experimentally measured strength. Recommendations and concluding remarks were outlined and discussed.

## 2. Experimental Database

An extensive experimental database for the experimental punching capacity of LWC slabs, which consists of 116 records, was collected [30,31,32,33,34,35,36,37,38,39,40,41,42,43,44,45]. Each record contains the following data: (1) concrete density (γ) in kN/m^3^; (2) column width (short side) (a) in m; (3) column length (long side) (b) in m, (for circular columns a=b=0.785 column diameter); (4) slab depth (d) in m; (5) 28-day cylinder compressive strength of concrete ($f_{c}^{\prime }$) in MPa; (5) longitudinal reinforcement ratio x yield strength of steel ($\mu f_{y}$) in MPa; (6) ultimate punching capacity ($V_{u}$) in kN. These collected records were divided into a training set (90 records) and a validation set (26 records). Table 1 summarizes their statistical characteristics. In addition, Figure 1 shows the histograms for both inputs and outputs.

## 3. Selected Design Model

In this section, selected design models are presented and compared with each other in terms of the various mechanisms and parameters considered, as shown in Table 2. In the present paper, we consider both EC2 and ACI.

For the EC2, the punching shear strength$\;\left(V_{u}\right)$ is calculated such that:(1)$$V_{u}=0.15\;\lambda \lambda_{s}\sqrt[{3}]{100\mu f\prime_{c}}\;\left(\frac{\pi }{2}\left[a+b+8d\right]\right)d\geq 0.028\;\lambda \lambda_{s}\sqrt[{3/2}]{f\prime_{c}}\left(\frac{\pi }{2}\left[a+b+8d\right]\right)d$$
where$\;\lambda_{s}=\left(1+\sqrt{\frac{200}{d}}\right)\leq 2.0,\;\mu \leq 0.02,$ λ=0.40+0.60 γ/2200, γ is the concrete density in kg/m^3^.

Whereas, for the ACI, $V_{u}$ is calculated such that:(2)$$V_{u}=0.17\left(1+\frac{2}{\beta_{0}}\right)\lambda^{\prime }\lambda_{s}{}^{\prime }\sqrt{f\prime_{c}}\;\left(2\left[a+b+2d\right]\right)d\leq 0.33\;\lambda^{\prime }\lambda_{s}{}^{\prime }\sqrt{f\prime_{c}}\;\left(2\left[a+b+2d\right]\right)$$
where $\;\lambda_{s}{}^{\prime }=\sqrt{\frac{2}{1+0.004d}}\leq 1$, $\lambda^{\prime }=3\frac{\gamma }{6400},$ $\beta_{0}\;$is the ratio between loading area dimensions.

## 4. Correlation and Effective Parameters

This section calculates the correlation matrix for all parameters to identify the dependent variables, as presented in Table 3. It is clear that the shear strength is highly correlated to the following parameters γ, a, b, d, $f_{c}^{\prime }$; however, it has a very low correlation to the parameter $\mu f_{y}$. It is worth noting that this conclusion violates the physically observed punching shear resisting mechanism for dowel action, which depends mainly on the flexure reinforcement ratio. This violation could be because the correlation coefficient is accurate for a linear relationship and is not as accurate for nonlinear ones. Thus, the AI models could provide a more accurate prediction for the significance of the parameters.

## 5. AI Model Development

Artificial intelligence (AI) techniques are searching algorithms that aim to find the best solution for certain problems according to certain criteria within the available time and resources. (AI) techniques may be classified into statistically based techniques (such as fuzzy logic), decision-tree-based techniques (such as expert systems), human brain simulation techniques (such as neural networks), evolutionary-based techniques (such as genetic algorithm and genetic programming), and mimicking natural creature behavior techniques (such as ant colony and particle swarm) [46].

ANN is a well-known technique that simulates the human brain activity. The model consists of a number of cells (nodes) arranged in groups (layers), where the cells of each group are connected to cells of other groups with links. Each link has its importance (weight) and each cell has its triggering mechanism (activation function). During the training stage, information propagates from the first group (input layer) to the last group (output layer) through the intermediate groups (hidden layers). The errors in the predicted values are backpropagated to the first group (input layer). During this process, the weights of links are updated to enhance the performance. ANN is a very powerful and flexible tool that can simulate any non-linear behavior, but its output is a weights matrix that cannot be utilized manually. Figure 2 presents a general schematic for the artificial neural network.

Genetic algorithms (GA) are AI-optimizing techniques that mimic the evolution procedure of living creatures. It starts with generating a random set of solutions, and then the fitness of each solution is evaluated, the most fitting solutions are selected, and the rest are deleted. Those selected solutions are used to produce the next generation of solutions, and the cycle continues until the desired accuracy is achieved. This technique was used as a base to develop more AI techniques, such as genetic programming (GP) and EPR.

GP is a technique that uses GA to optimize the fitness of the mathematical formula to a certain dataset. The output of this technique is a closed-form equation that could be utilized manually. EPR is another technique based on GA, where GA is used to optimize the traditional polynomial regression by selecting the most affecting terms and deleting the rest. The output of this technique is also a closed-form equation (polynomial) that could be utilized manually [47]. Figure 3 shows the mathematical formula in a tree and genetic form, whereas Figure 4 illustrates the flow chart of the EPR technique.

Three different artificial intelligent (AI) techniques were used to predict the ultimate punching capacity of LWC slabs using the collected database. These techniques are genetic programming (GP), artificial neural network (ANN), and polynomial regression optimized using a genetic algorithm, which is known as evolutionary polynomial regression (EPR). All of the three developed models were used to predict the ultimate punching capacity (***V_u_*** in kN) using the concrete density (γ in kN/m3), column width (a in m), column length (b in m), slab depth (d in m), 28-day concrete cylinder strength ($f_{c}^{\prime }$ in MPa), and reinforcement ratio by steel yield stress ($\mu f_{y}\;$in MPa).

Each model on the three developed models was based on a different approach (evolutionary approach for GP, mimicking biological neurons for ANN, and optimized mathematical regression technique for EPR). However, for all developed models, the prediction accuracy was evaluated in terms of the sum of squared errors (SSE), which is calculated such that:(3)$$\mathrm{SSE}=\overset{n}{\underset{i=1}{\sum }}{\left(y_{i}-f_{i}\right)}^{2}$$

The following section discusses the results of each model. The accuracies of the developed models were evaluated by comparing the (SSE) between predicted and calculated shear strength parameters values. The results of all developed models are summarized in Table 4.

### 5.1. GP Model

The developed GP model has four levels of complexity. The population size, survivor size, and number of generations were 100,000, 25,000, and 100, respectively. Equation (4) presents the output formulas for (***V_u_***), whereas Figure 5a shows its fitness. The average error % of total dataset is (32.4%), while the (R^2^) value is (0.823).
(4)$$V_{u}=58.5d\;\left(f_{c}^{\prime a}+\gamma +3\right)\left(11{\;d}^{2}f_{c}^{\prime b}+d+d\;\mu \mathrm{Fy}+1\right)$$

### 5.2. ANN Model

A backpropagation ANN with one hidden layer and nonlinear activation function (Hyper Tan) was used to predict (Vu) values. The used networks layout and their weights are illustrated in Figure 6 and Table 5. The average error % of total dataset was (26.1%) and the (R^2^) value was (0.890). The relative importance values for each input parameter are illustrated in Figure 7, which indicates that the slab thickness is the most important factor, whereas the concrete strength and flexure reinforcement ratio is second after the depth. The relationship between the calculated and predicted values are shown in Figure 5b.

### 5.3. EPR Model

Finally, the developed EPR model was limited to the quadrilateral level. For six inputs, there are 210 possible terms (126 + 56 + 21 + 6+1 = 210) as follows:(5)$$\overset{i=6}{\underset{i=1}{\sum }}\overset{j=6}{\underset{j=1}{\sum }}\overset{k=6}{\underset{k=1}{\sum }}\overset{l=6}{\underset{l=1}{\sum }}X_{i}\cdot X_{j}\cdot X_{k}\cdot X_{l}+\overset{i=6}{\underset{i=1}{\sum }}\overset{j=6}{\underset{j=1}{\sum }}\overset{k=6}{\underset{k=1}{\sum }}X_{i}\cdot X_{j}\cdot X_{k}+\overset{i=6}{\underset{i=1}{\sum }}\overset{j=6}{\underset{j=1}{\sum }}X_{i}\cdot X_{j}+\overset{i=6}{\underset{i=1}{\sum }}X_{i}+C$$

The GA technique was applied on these 210 terms to select the most effective ten terms to predict the values of (Vu). The outputs are illustrated in Equation (5), and its fitness is shown in Figure 7c. The average error % and (R^2^) values were (26.4%—0.888) for the total datasets.
(6)$$\begin{array}{c} V_{u}=\frac{200,000{\;a}^{2}\;b\left(a-155{\;b}^{2}d\right)+53.6{\;\gamma }^{2}\;b\;d}{a\;f_{c}^{\prime }}+\frac{\left(3165{\;a}^{2}-\gamma \right){\;d}^{3}f_{c}^{\prime }+7100{\;a}^{2}{\;b}^{4}}{\;3.7\;a\;b\;d\;}\\ +30,000{\;a\;b}^{2}d\left(700\;b\;d-700\;d^{2}-\;b\;f_{c}^{\prime }\right)-4.1 \end{array}$$

## 6. Safety of Proposed and Existing Models

The safety of the strength calculated using a specific model will be examined using the ratio between the measured strength and that calculated using that specific model (SF). Applying statistical measures on the SF calculated using the proposed models and existing ones could express the model’s accuracy, consistency, and safety. The closer the average of the SF to unity, the more accurate the model used. The lower the coefficient of variation of the SF, the more consistent the model. If the lower 95% confidence limit is close to unity and larger than 0.85, the model has acceptable safety. In addition, the SF can be plotted versus all effective parameters; thus, the variation of the model safety can be examined. In addition, the scattering for each effective parameter indicates the ability of the design model to accurately model the effect of each parameter, which varies significantly for each design model. Moreover, extreme values are observed for SF, which are considered statistical outliers and not an essential part of our conclusions. However, our analysis is based on statistical measures, as shown earlier in this section. This technique was implemented by several researchers [48].

### 6.1. Overall Safety of Various Models

Table 6 shows the statistical measures for the SF calculated using various methods. The three AI have less scattering than existing selected design codes. In addition, from the statistical measures, the three proposed AI models are more accurate, consistent, and reasonably safe than the existing design codes in terms of the mean, coefficient of variation, and lower 95%.

### 6.2. Safety of Various Models Versus Slab Size

Figure 8 shows the SF calculated using ANN, GP, EPR, ACI, and EC2 models versus the effective depth, as well as the best fit line for each method. The figure shows that the safety of ACI and EC2 increases with the decrease in the slabs size. On the other hand, the safety of the GP, ANN, and EPR methods was consistent concerning the effective depth. This is because the AI models captured the influence of the slabs size on the strength.

### 6.3. Safety of Various Models Versus Concrete Compressive Strength

Figure 9 shows the SF calculated using ANN, GP, EPR, ACI, and EC2 models versus the concrete strength, as well as the best fit line for each method. The figure shows that the safety of ACI and EC2 decreases with the increase in the concrete strength. On the other hand, the safety of the GP, ANN, and EPR methods was consistent with the concrete strength. This is because the AI models captured the influence of the concrete strength on the strength.

### 6.4. Safety of Various Models Versus Concrete Density

Figure 10 shows the SF calculated using ANN, GP, EPR, ACI, and EC2 models versus the concrete density, as well as the best fit line for each method. The figure shows that the safety of ACI and EC2 decreases with the increase in the concrete density. On the other hand, the safety of the GP, ANN, and EPR methods was consistent concerning the concrete density. This is because the AI models captured the influence of the concrete density on the strength.

### 6.5. Safety of Various Models Versus Column Dimension to Depth Ratio

Figure 11 shows the SF calculated using ANN, GP, EPR, ACI, and EC2 models versus the column dimensions, as well as the best fit line for each method. The figure shows that the safety of ACI and EC2 is different relative to the column dimensions. This is due to the fact that ACI considers the column dimension, whereas EC2 does not. On the other hand, the safety of the GP, ANN, and EPR methods was consistent concerning the column dimensions. This is because the AI models captured the influence of the column dimensions on the strength.

### 6.6. Safety of Various Models Versus Flexure Reinforcements

Figure 12 shows the SF calculated using the ANN, GP, EPR, ACI, and EC2 models versus the flexure reinforcement ratio, as well as the best fit line for each method. The figure shows that the safety of the ACI and EC2 is not consistent with the flexure reinforcement ratio compared to the proposed models. In addition, the ACI safety increases with the increase in the flexure reinforcement. On the other hand, the safety of the EC2 decreases with the increase in the flexure reinforcement ratio, which is because EC2 considers the effect of the flexure reinforcement ratio and EC2 does not. Moreover, the safety of the GP, ANN, and EPR methods was consistent concerning the flexure reinforcement ratio. This is because the AI models captured the influence of the flexure reinforcement ratio on the strength.

## 7. Future Studies

It is recommended that the following future studies be further investigated:

Design code development for cases of tension forces [48].Investigating punching shear with different tubes of columns [49,50];More machine learning methods [51,52];The behavior of full-scale slabs with thickness larger than 180 mm.The effect of using fibers in the concrete mix of lightweight concrete on the punching shear strength.

## 8. Conclusions

This research presents three models using three (AI) techniques (GP, ANN, and EPR) to predict the punching capacity of LWC slabs (***V_u_***) using the concrete density (γ), columns dimensions (a & b), slab depth (d), concrete strength ($f_{c}^{\prime }$), and reinforcement ratio by steel yield stress ($\mu f_{y}\;$). Although concluding remarks are limited to the range of parameter values in the database, which can improve with more testing of slabs, concluding remarks are as follows:

Both (ANN) and (EPR) have the greatest prediction accuracy (73.9% and 73.6%, respectively), whereas the (GP) model has the lowest prediction accuracy (67.6%);(GP) and (EPR) have almost the same level of accuracy (65.3% and 68.1%, respectively);Although the error% of the (ANN) and (EPR) models are so close, the output of (EPR) is closed-form equations, which could be used manually or as software, unlike the (ANN) output, which cannot be used manually;The summation of the absolute weights of each neuron in the input layer of the developed (ANN) model indicates that the slab depth (d) has a major influence on the punching capacity; other parameters have a minor effect, especially the compressive strength of the concrete;The formula developed using (EPR) did not include the parameter ($\mu f_{y}\;$), which indicates its minor effect on the punching capacity.The GA technique successfully reduced the 210 terms of the conventional polynomial regression quadrilateral formula to only ten terms without significant impact on its accuracy.AI models captured the true behavior and overcame the variability of the traditional design codes concerning the effective parameters.

## Figures and Tables

**Figure 1 materials-15-02732-f001:**
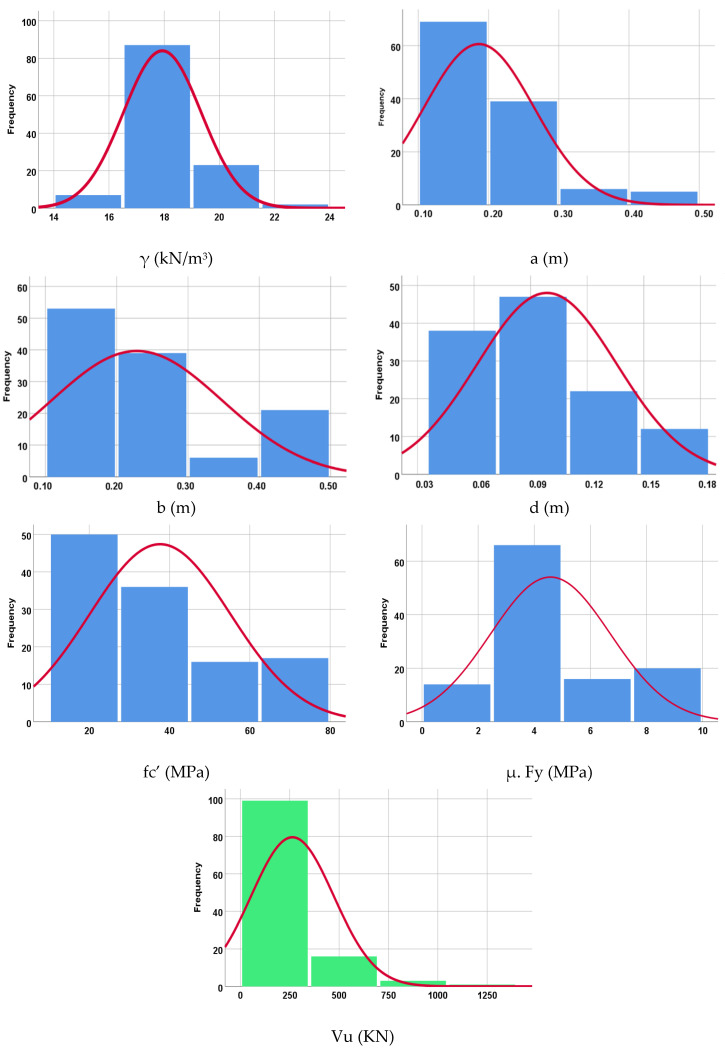
Distribution histograms for inputs (in blue) and outputs (in green).

**Figure 2 materials-15-02732-f002:**
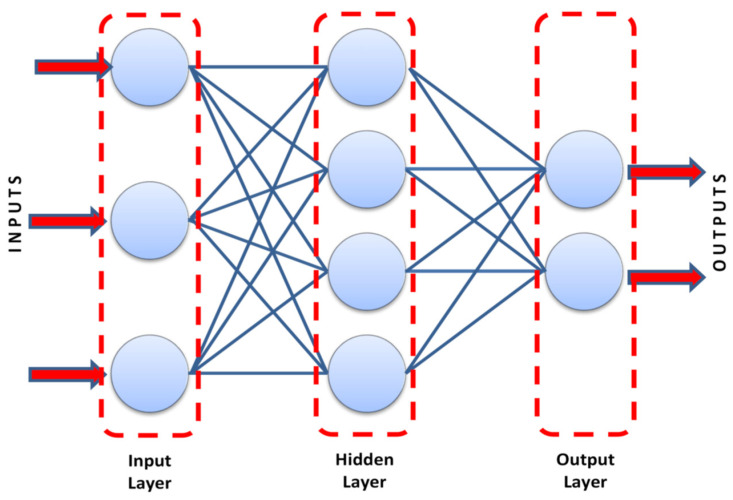
General schematic for the artificial neural network [47], reproduced with permission from Elsevier.

**Figure 3 materials-15-02732-f003:**
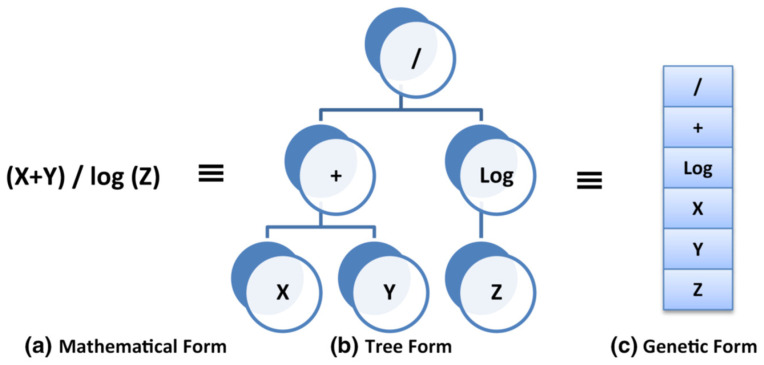
Presentation of mathematical formula in tree and genetic form [47], reproduced with permission from Elsevier.

**Figure 4 materials-15-02732-f004:**
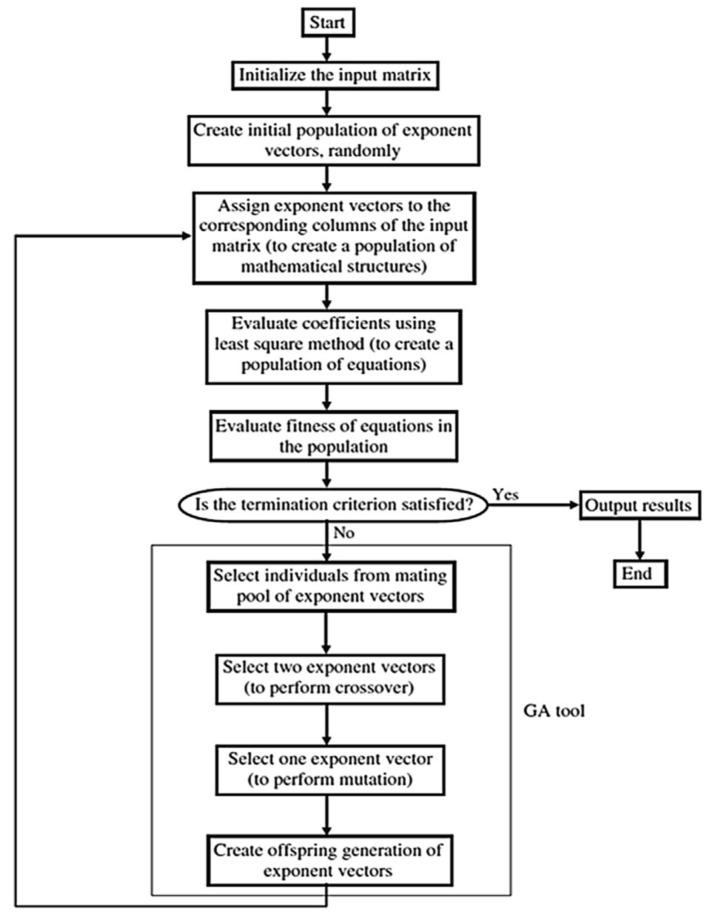
Typical flow diagram of the evolutionary polynomial regression (EPR) procedure [46], reproduced with permission from Springer Nature.

**Figure 5 materials-15-02732-f005:**
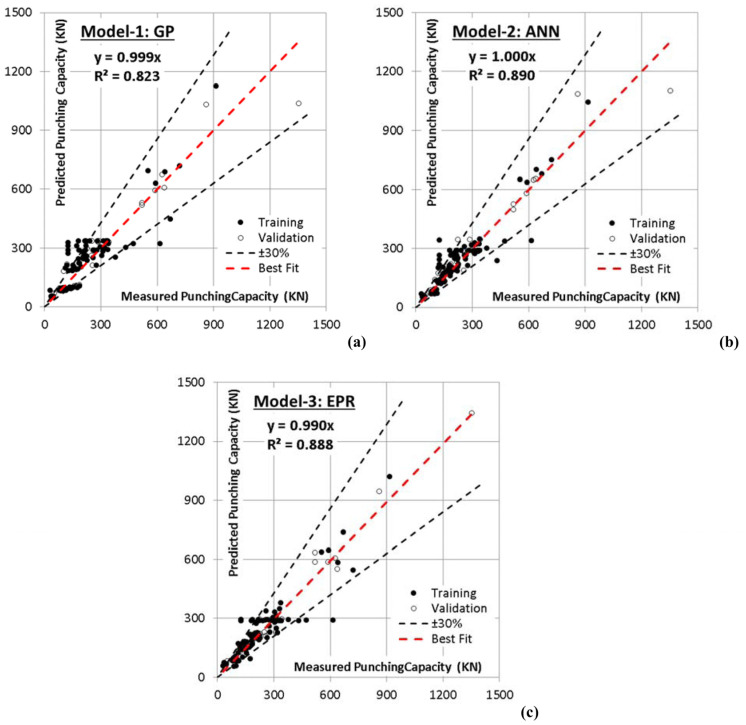
Relative relationship between predicted and calculated (***V_u_***) values using the developed models, (**a**) GP, (**b**) ANN, (**c**) EPR

**Figure 6 materials-15-02732-f006:**
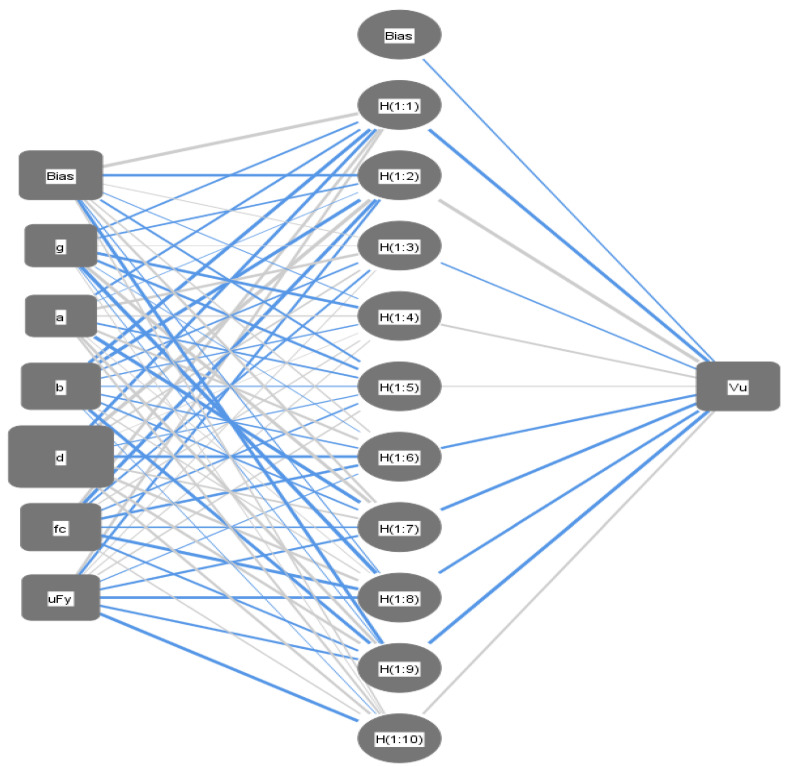
Layout for the developed ANN.

**Figure 7 materials-15-02732-f007:**
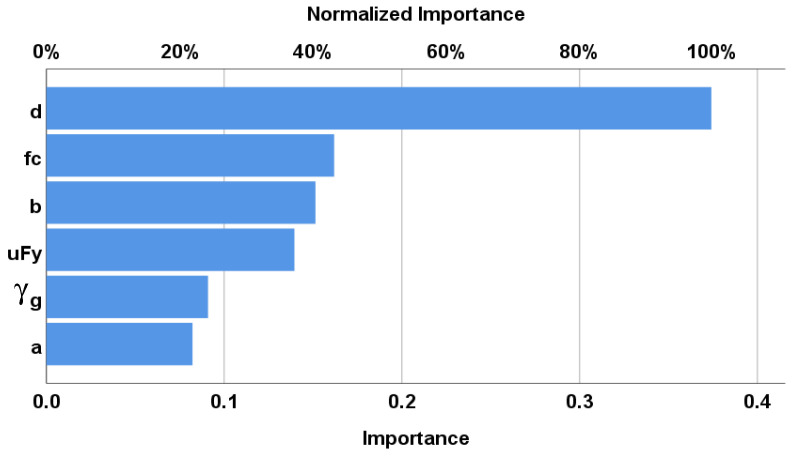
Relative importance of input parameters.

**Figure 8 materials-15-02732-f008:**
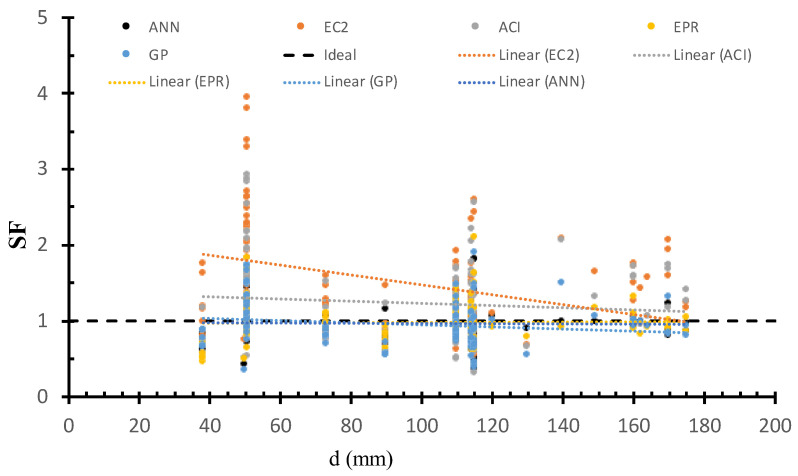
SF calculated using various models versus d.

**Figure 9 materials-15-02732-f009:**
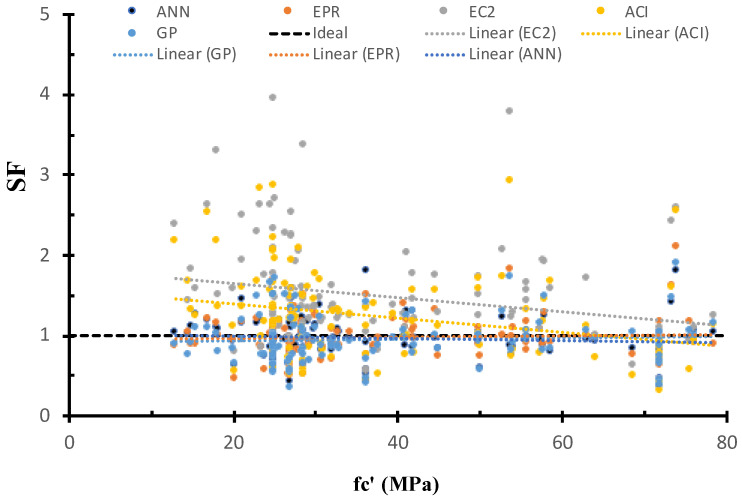
SF calculated using various models versus $f_{c}^{\prime }$.

**Figure 10 materials-15-02732-f010:**
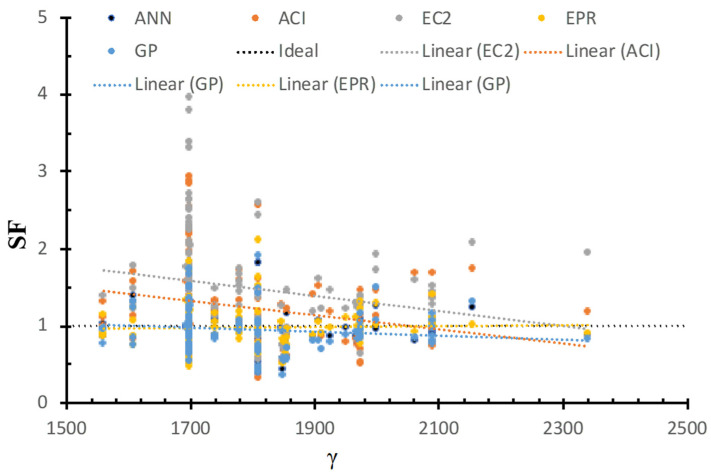
SF calculated using various models versus γ.

**Figure 11 materials-15-02732-f011:**
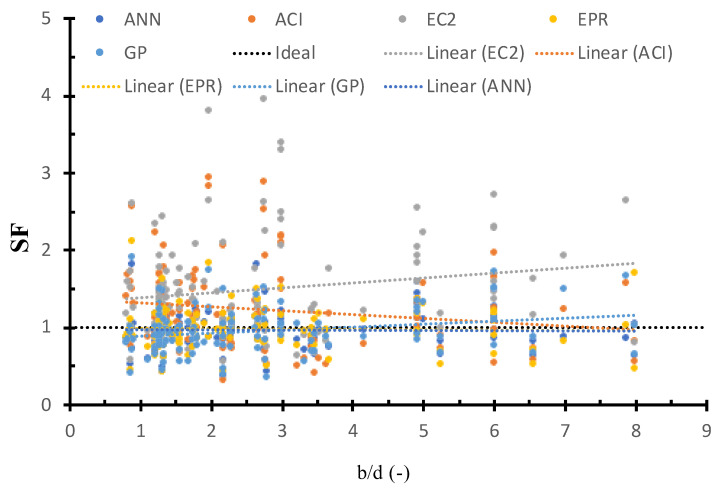
SF calculated using various models versus bd.

**Figure 12 materials-15-02732-f012:**
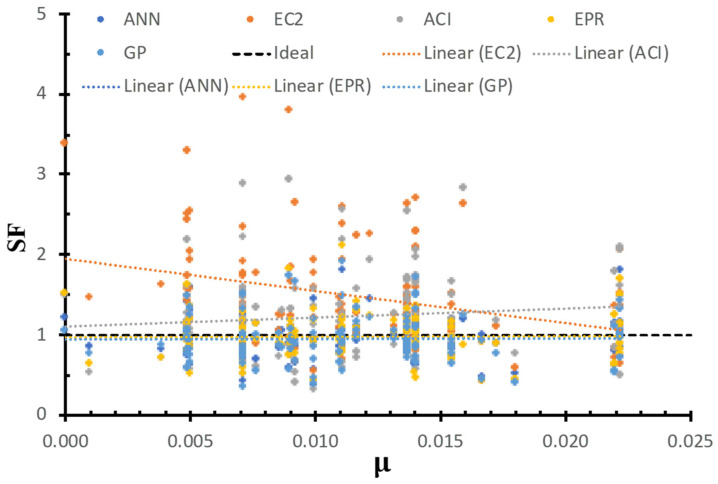
SF calculated using various models versus μ.

**Table 1 materials-15-02732-t001:** Statistical analysis of collected database.

	γ	a	b	d	$f_{c}^{\prime }$	$\mu f_{y}$	$V_{u}$
	kN/m^3^	m	m	m	MPa	MPa	kN
Training set							
Min.	15.60	0.10	0.10	0.04	12.96	1.05	29.00
Max.	23.40	0.40	0.46	0.18	78.40	9.48	914.00
Avg.	18.00	0.19	0.22	0.10	37.98	4.52	245.38
SD	1.38	0.07	0.11	0.04	18.43	2.13	181.90
VAR	0.08	0.39	0.49	0.37	0.49	0.47	0.74
Validation set							
Min.	15.60	0.11	0.11	0.04	21.10	0.00	46.59
Max.	21.56	0.41	0.46	0.18	72.00	8.56	1354.00
Avg.	17.79	0.19	0.24	0.10	37.93	4.79	282.99
SD	1.51	0.09	0.13	0.04	14.08	2.11	256.77
VAR	0.08	0.49	0.56	0.39	0.37	0.44	0.91

**Table 2 materials-15-02732-t002:** Comparison between the parameters included by each of the selected models.

Mechanism	EC2	ACI
Friction across crack in terms of $f_{c}^{\prime }$.	√	√
Dowel action mechanism in terms of μ.	√	×
Concrete type in terms of γ.	√	√
Column dimension in terms of a, b	×	×
Direct shear mechanism in terms of compression zone depth in the strength.	×	×
Size effect in terms of d.	√	√
Aggregate interlock mechanism in terms of aggregate size and type.	×	×
Arch action mechanism in terms of shear span to depth ratio.	×	×
Flexure capacity of the slab cross section.	×	×

**Table 3 materials-15-02732-t003:** Pearson correlation matrix.

	γ	a	b	d	$f_{c}^{\prime }$	$\mu f_{y}$	$V_{u}$
Γ	1.00						
a	0.07	1.00					
b	−0.15	0.55	1.00				
d	0.34	0.21	0.30	1.00			
$f_{c}^{\prime }$	0.39	0.28	−0.03	0.44	1.00		
$\mu f_{y}$	−0.02	0.15	0.15	−0.15	−0.16	1.00	
$V_{u}$	0.38	0.46	0.39	0.78	0.40	0.05	1.00

**Table 4 materials-15-02732-t004:** Accuracies of developed models.

Technique	Model	SSE	Avg. Error %	R^2^
GP	Equation (1)	780,494	32.4	0.823
ANN	Figure 2	506,732	26.1	0.890
EPR	Equation (2)	518,119	26.4	0.888

**Table 5 materials-15-02732-t005:** Weights matrix for the developed ANN model.

		**Hidden Layer**										
		H (1:1)	H (1:2)	H (1:3)	H (1:4)	H (1:5)	H (1:6)	H (1:7)	H (1:8)	H (1:9)	H (1:10)	
Input Layer	(Bias)	1.10	−0.50	0.07	−0.09	−0.23	0.14	0.26	−0.12	−0.53	0.15	
	γ	−0.22	−0.19	0.02	−0.51	−0.38	−0.04	0.68	−1.42	−0.06	0.05	
	a	−0.24	−0.03	0.47	0.14	−0.19	0.35	−0.84	0.05	0.43	0.23	
	b	−0.79	−0.81	−0.18	−0.14	−0.13	−0.16	−0.19	0.13	−0.67	−0.03	
	d	−0.51	1.57	0.16	0.01	−0.13	−0.52	0.22	0.43	0.38	0.33	
	$f_{c}^{\prime }$	0.40	−0.73	−0.19	0.13	−0.16	−0.47	−0.16	−0.75	−0.24	0.14	
	$\mu f_{y}$	0.33	−0.30	0.07	0.09	0.15	−0.11	−0.29	−0.59	−0.32	−0.71	
		Hidden Layer										
Output layer		H (1:1)	H (1:2)	H (1:3)	H (1:4)	H (1:5)	H (1:6)	H (1:7)	H (1:8)	H (1:9)	H (1:10)	(Bias)
	$V_{u}$	−0.15	−0.69	0.82	−0.15	0.23	0.13	−0.36	−0.67	−0.48	−0.79	0.26

**Table 6 materials-15-02732-t006:** Statistical methods for SF using various models.

	**GP**	**ANN**	**EPR**	**ACI**	**EC2**
Maximum	1.81	1.90	2.10	2.93	3.94
Minimum	0.36	0.35	0.42	0.31	0.43
Average	0.97	0.95	0.98	1.24	1.49
C.O.V.	25%	31%	29%	43%	44%
Lower 95%	0.92	0.9	0.93	1.14	1.37

## Data Availability

All data are available within the manuscript.

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
