# Peer review of "Using Artificial Intelligence Techniques to Predict Punching Shear Capacity of Lightweight Concrete Slabs"

_materials, 2022, doi:10.3390/ma15082732_

Round 1

Reviewer 1 Report

The paper presents a study of Using Artificial Intelligence Techniques to Predict Punching Shear Capacity of Lightweight Concrete Slabs. It is an interesting topic to the researchers in the related areas.  However, the work would be excellent if some mistakes were  revised and some explanation were given.

1.The author can describe more about the contribution of the article, what is the innovation of this paper. 

2.In Section 7 CONCLUSIONS, future research can be described in the paper.

Author Response

Reviewer 1

Comments and Suggestions for Authors

The paper presents a study of Using Artificial Intelligence Techniques to Predict Punching Shear Capacity of Lightweight Concrete Slabs. It is an interesting topic to the researchers in the related areas. However, the work would be excellent if some mistakes were revised, and some explanations were given.

  1. The author can describe more about the contribution of the article, what is the innovation of this paper.

For clarity, please see the abstract where it states that “The manuscript presented several artificial intelligence models that can be used to better predict the punching shear strength of lightweight concrete slabs. In addition, it provided guidance for future design code development, where the importance of each variable on the strength was identified. Moreover, it provided an expression showing the complicated inter-relation between affective variables. The novelty lies in developing three innovative prediction models for the punching capacity of lightweight concrete slabs using three different (AI) techniques capable of accurately predicting the strength compared to the experimental database.

  1. In Section 7 CONCLUSIONS, future research can be described in the paper.

For clarity, please see conclusion where it states that “It is recommended that following future studies be further investigated:

  • More machine learning methods.
  • Design code development.
  • Behavior of full-scale slabs with thickness larger than 180 mm.
  • The effect of using fibers in the concrete mix of lightweight concrete on the punching shear strength.”

Reviewer 2 Report

This research presents three models using three (AI) techniques (Genetic programming, Artificial neural networks, Evolutionary Polynomial Regression) to predict the punching capacity of to predict the punching capacity of lightweight concrete slabs using concrete density, columns dimensions, slab depth, concrete strength, reinforcement ratio by steel yield stress. However, there are still some problems to be improved, specific comments are as follows:

  1. What is the innovation of this manuscript? The authors should strengthen it.
  2. For abbreviations in this manuscript, the full spelling is given on the first use, check the full text, such as LWC, ACI, et. al.
  3. The description of the abstract is not good enough and does not capture the core or focus of the manuscript. For example, the three machine learning methods in this paper should be given in the abstract instead of just mentioning the three machine learning methods used and need to be revised.
  4. The form of Eq. 1 and 2 is quite confusing and misunderstood, refine them.
  5. Section 5 should add schematics to the various methods.
  6. In Section 5, the formula for Sum of Squared Errors needs to be given.
  7. The manuscript is poor in content and presentation and requires full text revision.
  8. The analysis of the results of the sixth section is very confusing and the content is not refined enough.
  9. In Section 5, when mentioning the AI techniques used for model development, more recent relative progress should be strengthened, like “Computer Methods in Applied Mechanics and Engineering, 2022, 388: 114218”.
  10. In Section 6, the order of the pictures should match the order of the text in the manuscript. i.e. the order in which each method appears.
  11. Reduce the number of pictures in the manuscript, especially Section 6, reduce the size of the pictures, and consolidate the pictures.
  12. Check the X-axis coordinates of Figure 8.
  13. The accuracy of the model is very low and has little reference value. The results are not enough to support a paper.
  14. The English usage needs to be fully refined for further evaluation.

Author Response

Reviewer 2

Comments and Suggestions for Authors

This research presents three models using three (AI) techniques (Genetic programming, Artificial neural networks, Evolutionary Polynomial Regression) to predict the punching capacity of to predict the punching capacity of lightweight concrete slabs using concrete density, columns dimensions, slab depth, concrete strength, reinforcement ratio by steel yield stress. However, there are still some problems to be improved, specific comments are as follows:

  1. What is the innovation of this manuscript? The authors should strengthen it.

For clarity, please see the abstract where it states that “The manuscript presented several artificial intelligence models that can be used to better predict the punching shear strength of lightweight concrete slabs. In addition, it provided guidance for future design code development, where the importance of each variable on the strength was identified. Moreover, it provided an expression showing the complicated inter-relation between affective variables. The novelty lies in developing three innovative prediction models for the punching capacity of lightweight concrete slabs using three different (AI) techniques capable of accurately predicting the strength compared to the experimental database.

  1. For abbreviations in this manuscript, the full spelling is given on the first use, check the full text, such as LWC, ACI, et. al.

For clarity, please see L

  1. The description of the abstract is not good enough and does not capture the core or focus of the manuscript. For example, the three machine learning methods in this paper should be given in the abstract instead of just mentioning the three machine learning methods used and need to be revised.

Done, please see abstract.

  1. The form of Eq. 1 and 2 is quite confusing and misunderstood, refine them.

Done, please see equation 1 and 2.

  1. Section 5 should add schematics to the various methods.

Done, please see added figures.

  1. In Section 5, the formula for Sum of Squared Errors needs to be given.

Done, please see section 5.

  1. The manuscript is poor in content and presentation and requires full-text revision.

For clarity, the manuscript was revised thoroughly.

  1. The analysis of the results of the sixth section is very confusing and the content is not refined enough.

Done, please see section 6.

  1. In Section 5, when mentioning the AI techniques used for model development, more recent relative progress should be strengthened, like “Changqi Luo Behrooz Keshtegar Shun Peng Zhu Osman Tayl and Xiao-Peng Niu. Hybrid enhanced Monte Carlo simulation coupled with advanced machine learning approach for accurate and efficient structural reliability analysis. Computer Methods in Applied Mechanics and Engineering, 2022, 388: 114218”.

Done

  1. In Section 6, the order of the pictures should match the order of the text in the manuscript. i.e. the order in which each method appears.

Done see figs in section 6.

  1. Reduce the number of pictures in the manuscript, especially Section 6, reduce the size of the pictures, and consolidate the pictures.

Done see figs in section6.

  1. Check the X-axis coordinates of Figure 8.

Done.

  1. The accuracy of the model is very low and has little reference value. The results are not enough to support a paper.

Fig. 5 presented a comparison between the three developed models and the provisions of EC2 and ACI, which showed that the predicted values using (AI) techniques are much more accurate than current empirical formulas.

  1. The English usage needs to be fully refined for further evaluation.

For clarity, the manuscript was proofread by several native language colleges and corrections were implemented.

Reviewer 3 Report

This paper investigates the prediction of the punching shear strength of lightweight concrete slabs using AI technique. The paper can be accepted subjected to minor revision. 1. Can you mention the ratio of the training and validation ratio of the data set? 2. Some important recent research of the AI technique in structural engineering is missing in the literature review section. The following papers are suggested to discuss in the literature review section: (a) Ahmed M, Tran V-L, Ci J, Yan X-F, and Wang F. Computational analysis of axially loaded thin-walled rectangular concrete-filled stainless steel tubular short columns incorporating local buckling effects. Structures 2021;34:4652-4668. (b) Ci J, Ahmed M, Tran V-L, Jia H, and Chen S. Axial compressive behavior of circular concrete-filled double steel tubular short columns. Advances in Structural Engineering 2021:13694332211046345. (c) Tran V-L and Kim S-E. A practical ANN model for predicting the PSS of two-way reinforced concrete slabs. Engineering with Computers 2021;37:2303-2327. 3. In Fig. 9, the symbol for density in the horizontal axis is not showing correctly and need to edit it. 4. You need to provide the values of Mean, COV and Lower 95% for Figs. 6,7,8 (similar to Fig. 5) to make your claim (in section 6.2-6.4 about the safety of the models) justified. 5. Further discussions of the effects of each parameter on the design accuracy of design codes should be included e.g. why for a particular parameter, the safety of a particular design code either becomes conservative or over-conservative. The current discussion is very general. 6. To a surprise, I wonder why EC2 results in a very large ratio (SF) of the measured value vs predicted value? Some of the predictions are extremely high, as high as 8. Please check the calculation and provide a deep discussion on this context. You may also refer to some references where similar results were obtained. 7. I did not find where and how did you get this conclusion from: “GP technique successfully reduced the 210 terms of conventional polynomial regression quadrilateral formula to only ten terms without significant impact on its accuracy.”

Author Response

Reviewer 3

Comments and Suggestions for Authors

This paper investigates the prediction of the punching shear strength of lightweight concrete slabs using AI technique. The paper can be accepted subjected to minor revision.

Thank you appreciated

  1. Can you mention the ratio of the training and validation ratio of the data set?

For clarity, please see L80 where it states that “These collected records were divided into a training set (90 records) and validation set (26 records).”

  1. Some important recent research of the AI technique in structural engineering is missing in the literature review section. The following papers are suggested to discuss in the literature review section:

(a) Ahmed M, Tran V-L, Ci J, Yan X-F, and Wang F. Computational analysis of axially loaded thin-walled rectangular concrete-filled stainless steel tubular short columns incorporating local buckling effects. Structures 2021; 34:4652-4668.

(b) Ci J, Ahmed M, Tran V-L, Jia H, and Chen S. Axial compressive behavior of circular concrete-filled double steel tubular short columns. Advances in Structural Engineering 2021:13694332211046345.

(c) Tran V-L and Kim S-E. A practical ANN model for predicting the PSS of two-way reinforced concrete slabs. Engineering with Computers 2021; 37:2303-2327.

Done, please L236-237.

  1. In Fig. 9, the symbol for density in the horizontal axis is not showing correctly and need to edit it.

Done, see fig. 9

4.You need to provide the values of Mean, COV and Lower 95% for Figs. 6,7,8 (like Fig. 5) to make your claim (in section 6.2-6.4 about the safety of the models) justified.

Done.

  1. Further discussions of the effects of each parameter on the design accuracy of design codes should be included e.g. why for a particular parameter, the safety of a particular design code either becomes conservative or over-conservative. The current discussion is very general.

For clarity, please see L 189 where it states that “Scattering for each effective parameter indicates the ability of design model to accurately model the effect of parameter, which varies significantly for each design model”

And see L 227, where it states, “Which is because the AI models captured the influence of the concrete strength on the strength.”

And see L 242, where it states that “The figure shows that the safety of ACI and EC2 is different relative to the column dimensions. This is due to that ACI consider the column dimension while EC2 do not.”

And see L 252, where it states that “The figure shows that the safety of the ACI and EC2 is not consistent with the flexure reinforcement ratio compared to the proposed models. In addition, the ACI safety increases with the increase in the flexure reinforcement. On the other hand, the safety of the EC2 decreases with the increase in the flexure reinforcement ratio, which is because EC2 consider the effect of flexure reinforcement ratio and EC2 do not”.

  1. To a surprise, I wonder why EC2 results in a very large ratio (SF) of the measured value vs predicted value? Some of the predictions are extremely high, as high as 8. Please check the calculation and provide a deep discussion on this context. You may also refer to some references where similar results were obtained.

For clarity, please see L 204, where it states that “Moreover, extreme values are observed for SF which are considered statistical outliers and not an essential part of our conclusions. However, our analysis is based on statistical measures as shown earlier in this section.”

  1. I did not find where and how did you get this conclusion from: “GP technique successfully reduced the 210 terms of conventional polynomial regression quadrilateral formula to only ten terms without significant impact on its accuracy.”

For clarity, Please see it is a typing mistake, the correct one is “GA technique successfully reduced…” , please refer to section 5.2 for more details.

Round 2

Reviewer 2 Report

The work has been well refined, it can be accepted as it is. 

Reviewer 3 Report

The paper can be accepted. However, just a small correction required during the proofreading of the paper that is the symbol for density still does not show correctly in Fig. 10. Please check.